# Enhancing Electroretinogram Classification with Multi-Wavelet Analysis and Visual Transformer

**DOI:** 10.3390/s23218727

**Published:** 2023-10-26

**Authors:** Mikhail Kulyabin, Aleksei Zhdanov, Anton Dolganov, Mikhail Ronkin, Vasilii Borisov, Andreas Maier

**Affiliations:** 1Pattern Recognition Lab, Friedrich-Alexander-Universität Erlangen-Nürnberg, 91058 Erlangen, Germany; andreas.maier@fau.de; 2Engineering School of Information Technologies, Telecommunications and Control Systems, Ural Federal University Named after the First President of Russia B. N. Yeltsin, 620002 Yekaterinburg, Russia; a.e.zhdanov@urfu.ru (A.Z.); anton.dolganov@urfu.ru (A.D.); m.v.ronkin@urfu.ru (M.R.); v.i.borisov@urfu.ru (V.B.)

**Keywords:** biomedical research, classification, deep learning, wavelet analysis, electroretinography, electroretinogram, ERG

## Abstract

The electroretinogram (ERG) is a clinical test that records the retina’s electrical response to light. Analysis of the ERG signal offers a promising way to study different retinal diseases and disorders. Machine learning-based methods are expected to play a pivotal role in achieving the goals of retinal diagnostics and treatment control. This study aims to improve the classification accuracy of the previous work using the combination of three optimal mother wavelet functions. We apply Continuous Wavelet Transform (CWT) on a dataset of mixed pediatric and adult ERG signals and show the possibility of simultaneous analysis of the signals. The modern Visual Transformer-based architectures are tested on a time-frequency representation of the signals. The method provides 88% classification accuracy for Maximum 2.0 ERG, 85% for Scotopic 2.0, and 91% for Photopic 2.0 protocols, which on average improves the result by 7.6% compared to previous work.

## 1. Introduction

The electroretinogram (ERG) technique has tremendous potential for early disease detection, diagnosis, and interventions in the field of ophthalmology. The ERG signal is an electrophysiological signal that represents the retina’s electrical response [1]. In ophthalmology, ERG testing can be a valuable tool because it is noninvasive and relatively simple [2]. The significance of ERG research lies in its ability to understand better how the retina works and make identifying and tracking diseases easier.

Manual ERG analysis is highly dependent on the clinician’s experience and other human factors, as a misdiagnosis might mean that the patient misses the optimal time for treatment [3]. On the other hand, automated ERG signals analysis uses machine learning (ML) methods [4]. It is a so-called data-based approach that requires a large amount of data. ML algorithms allow us to diagnose certain diseases or conditions based on the retinal activity patterns detected in ERG data. It is believed that this will assist clinicians in making more accurate diagnoses and developing more effective treatment plans [5]. An ML algorithm can use the analysis of large datasets of ERG data to identify patterns and relationships between variables that can be used to predict disease progression, treatment response, and other outcomes [6]. Developing new treatment options is another application of ML that is becoming increasingly important in ERG research. In order to identify specific patterns of retinal activity associated with a positive response to treatment, ML algorithms can analyze ERG data from patients who have responded well to specific treatments. With the help of this information, new treatments can be developed that are specifically targeted toward these particular patterns of cellular activity [4,7,8].

Illustrations of the ERG signals of healthy and unhealthy subjects, along with the designation of the parameters that clinicians analyze, are shown in Figure 1. The clinician parameters of the ERG waveform, the amplitudes (Va, Vb) and latency of the so-called a-wave and b-wave (Ta, Tb), are leveraged to identify abnormalities and diagnose a range of retinal disorders [9,10,11,12,13].

Figure 1 shows the temporal representation of the ERG signal. Both cases for healthy and unhealthy typically exhibit distinct and recognizable waveforms. However, the shape of the ERG signal in temporal representation can vary depending on the underlying pathology [14]. Severe dysfunction or loss of photoreceptor and bipolar cell activity can result in significant reduction or absence of the a-wave and b-wave in certain cases [15]. A severe form of macular degeneration or advanced retinitis pigmentosa can result in this type of vision loss.

The ERG waveform may be selectively affected by certain diseases rather than the whole waveform. There is an indication that the b-wave may be reduced or absent in some cases of congenital stationary night blindness, whereas the a-wave remains relatively normal, indicating a defect in bipolar cell function [16,17].

Consequently, ERG signals may provide useful information about retinal cells’ integrity and function and aid in the diagnosis of various retinal diseases and disorders. An effective method of obtaining disease information is to search for the most presentable data representation in the database [18]. To avoid reliability issues, extensive efforts must be made to extract and select features. In order to achieve this, it is necessary to first search for informative representations of data and then apply clustering to the new features to select them.

As was shown in [7,19,20], the wavelet representation of ERG signals allows one to obtain highly reliable features to increase the accuracy of automated doctor assistance. During these studies, it was noticed that some of the tested mother wavelet selections may lead to slightly different representations of ERG signals. Thus, it can be assumed that using each wavelet decomposition approach may restrict the number of features and extracted information. Then, searching for the best wavelet combination can be suggested to overcome this problem. The suggested approach can be thought of as some ensemble at the preprocessing stage. Let us also note that the previous paper’s analysis shows that most of the research in the field has proposed different wavelets as optimal for different cases [20]. Thus, the proposed idea can be a suggestion of some generalized system.

This paper aims to investigate the best combination of deep learning (DL) models with images of wavelet scalograms and their combination (stack) as input. The paper’s contribution consists of showing the benefits of wavelet combination as input to the classifier of ERG signals. In order to address the applications, a decision method based on wavelet combinations and an architecture of DL convolution neural networks is proposed and tested. For the collected and balanced database, the possibility of simultaneous analysis of adult and pediatric signals is also shown.

## 2. Related Works

Nowadays, studies that explore the potential of artificial intelligence algorithms for accurately classifying eye diseases and recognize their role as supportive tools for medical specialists are becoming more widespread. For physicians to play a vital role in delivering comprehensive and holistic medical care, they must realize the complexity of human health, the significance of empathetic care, and their unmatched decision-making abilities.

In medical practice, the conventional manual ERG analysis is based on a four-component evaluation [9,10,11,12,13,21]. In some cases, Discrete Wavelet Transform (DWT) is also applied. That provides more accurate signal descriptions than time-domain data [22]. For instance, according to the study results [23], waveforms of transient pattern electroretinograms (PERGs) are more easily separated when they are represented as DWT coefficients for full-time domain signals rather than in traditional peak-based feature spaces based on peak detection.

Similar wavelet-based methods were leveraged to evaluate the ERG waveform in autistic spectrum disorder (ASD) and attention deficit hyperactivity disorder (ADHD) [3]. ERG analysis has been demonstrated to be more comprehensive when using the continuous wavelet transform instead of the conventional conventional approaches. The Morlet wavelet transform was suggested in [19] to quantify the frequency, peak time, and power spectrum of the oscillatory potentials components of the adults’ ERG, which provided more information than did other wavelet transforms used earlier in the study. In [24], the Gaussian wavelet was chosen for its convenience in semi-automatic parameter extraction for pediatric and adult ERGs and for its superior time domain properties.

The study [25] compares mother wavelets to analyze normal adults’ ERG waveforms by minimizing scatter in the results. The use of this approach improved the data analysis and level of accuracy. The study demonstrated that different wavelets emphasize different signal features, making choosing the most appropriate mother wavelet crucial. In [26], researchers conducted a preliminary analysis and found that ERG waveforms shaped by Ricker were the best matched to their expected waveforms. In work [27], the Morlet wavelet was also suggested for adult ERG analysis.

The paper [28] shows the Ricker wavelet exhibiting superior median accuracy values for ERG wavelet scalogram classification, potentially due to several factors. The distinctive characteristics of the Ricker wavelet, including its shape and frequency attributes, align favorably with the features observed in ERG wavelet scalograms. As a result, using the Ricker wavelet leads to improved classification accuracy compared to other wavelet types. This enhanced accuracy can be attributed to the wavelet’s superior time-frequency localization properties, which enhance its ability to differentiate between various ERG responses. According to the mentioned articles, the classification problem was successfully addressed, and frequency pattern estimates for ERGs were presented [18]. However, the problem of best wavelet selection has not been solved yet.

As shown above, the selection of an appropriate wavelet for ERG signal analysis depends on the waveform’s characteristics. Different wavelets exhibit varying frequencies and temporal resolutions. It can be assumed that for the achievement of accurate results, an optimal wavelet should possess effective noise suppression capabilities, accurately capture both transient and sustained components of the ERG signal, and provide interpretable coefficients for feature identification [26]. Computational efficiency is crucial for handling large datasets and real-time applications. Furthermore, the expertise of the researcher or clinician in interpreting specific wavelets plays a significant role in enhancing accuracy and efficiency. Therefore, careful wavelet selection is essential to ensure reliable and meaningful results in both clinical and research settings [25].

The ERG analysis based on only four parameters may be insufficient for precise diagnosis. Then, augmenting the feature space through continuous wavelet transform in the frequency-time domain becomes imperative. By incorporating this approach, the classification of ERG responses can be enhanced by capturing additional information encoded in the frequency-time characteristics of the signal [29]. In the Transformer model, for instance, the accuracy distribution is wide [30]. Even so, as the training dataset grows, this variability will decrease. Furthermore, testing data must be divided and preserved according to the distribution observed in real-world scenarios without modification. This division affected the quantity of available training data.

## 3. Dataset Investigation

The original dataset consists of 1975 signals acquired from 323 patients, encompassing both adults and children [31]. The signals comprise five distinct types: Scotopic 2.0 ERG response, Photopic 2.0 ERG response, Maximum 2.0 ERG response, Photopic 2.0 EGR Flicker response, and Scotopic 2.0 ERG Oscillatory Potentials. This investigation primarily focuses on the utilization and detailed analysis of the Scotopic 2.0 ERG response, Photopic 2.0 ERG response, and Maximum 2.0 ERG response as described in a comprehensive study [32], which includes statistical examination. The dataset was obtained through electrophysiological studies conducted at the IRTC Eye Microsurgery Yekaterinburg Center utilizing the EP-1000 computerized electrophysiological workstation developed by Tomey GmbH, Nuremberg, Germany. The Tomey EP-1000 is a medical device for performing electrophysiological tests and incorporates an integrated database for storing patient data. However, the Tomey EP-1000 does not enable easy access to test results. To extract the data from the Tomey EP-1000, specialized software [33] was employed.

The t-SNE-based visualization of the utilized dataset is shown in Figure 2. Figure 2a shows a visualization of three types of signals represented by different colors: blue for Maximum ERG Response, red for Scotopic ERG Response, and gray for Photopic ERG Response. Figure 2b shows healthy and unhealthy subjects, with healthy subjects represented by blue and unhealthy subjects represented by red.

The results in Figure 2b show that the adult and pediatric signals could be considered to be processed together due to the high mixing among them in each signal type. According to the distribution shown in Figure 2, the intragroup scatter of parameters matches the intergroup scatter between pediatric and adults. As a result of this reasoning, it is possible to conduct a joint analysis of healthy and unhealthy subjects belonging to different age groups.

The collected dataset shows the high unbalancing of the data classes. The balancing was performed using an under-sampling approach. The under-sampling was employed using the AllKNN function from the Imbalanced-learn package AIIKNN [34]. The AllKNN function employs the nearest neighbor algorithm to detect instances that exhibit inconsistencies within their local neighborhood.

In our study, we utilized the classical significant features of ERG signals as input for this function. AllKNN method has a hyperparameter that affects the results of the under-sampling procedure: setting it too low or too high could lead to either removing too much of the data or removing too few data. The goal of the under-sampling in this study is to ensure a balance between healthy and unhealthy groups. For that, an array of possible numbers was selected. In this case, for Maximum and Photopic signals, we have chosen empirically to use 13 as the number of nearest neighbors to achieve the desired class balance. It is worth mentioning that the Scotopic signals were inherently balanced and did not necessitate any under-sampling technique to maintain class equilibrium.

Table 1 presents the distribution of healthy and unhealthy subjects within a balanced dataset. In this work, we balanced the dataset for the training experiments. For the testing, we keep the “real-world scenario” distribution of the healthy and unhealthy patients, as the number of patients with eye diseases is always higher on the clinic tests.

## 4. Methods

### 4.1. Experiment

Figure 3 shows the pipeline of the experiments. In the study, a five-fold cross-validation approach was applied to assess the performance of the proposed methodology. Within this process, the test subset was segregated based on the actual distributions observed in clinical patients who were classified as healthy or unhealthy for each type of ERG response. Initially, CWT transform was applied. Subsequently, the remaining shuffled training subset was divided into five folds. One fold was assigned for validation, while the remaining four folds were utilized for training. This cycle was repeated five times, ensuring that each fold was used for the validation set once.

ADAM optimization with an initial learning rate of 0.001 was employed during training. Each model was trained until convergence using early stopping criteria based on the validation loss. A batch size of 16 was used, and training was performed on a single NVIDIA V100 graphics processing unit on a machine with two Intel Xeon Gold 6134 3.2 GHz and 96 GB RAM. The commonly utilized Cross-entropy loss function for classification tasks was employed to train the network models.

Data augmentation techniques were employed to augment the dataset. Specifically, geometric transformations such as random cropping, vertical flipping, and image translation were exclusively utilized on the images under consideration.

### 4.2. Continuous Wavelet Transform

Continuous Wavelet Transform (CWT) stands as a potent mathematical instrument that provides an overcomplete representation of a signal by letting the translation and scale parameter of the wavelets vary continuously. CWT of a function x(t) at *a* scale $\left(a>0\right)\in R^{+*}$ and translational value b∈R is expressed by the following integral (Equation 1), where ψ(t) is a continuous function called the mother wavelet, and the overline represents the operation of complex conjugate [35]. The primary objective of the mother wavelet is to serve as a foundational function for generating daughter wavelets, which are simply the translated and scaled versions of the mother wavelet. The output of the CWT consists of a two-dimensional time-scale representation of the signal.
(1)$$X_{w}\left(a,b\right)=\frac{1}{{\left|a\right|}^{1/2}}\int_{-\infty }^{\infty }x\left(t\right)\bar{\psi }\left(\frac{t-b}{a}\right)dt$$

The wavelet transformation was carried out using PyWavelets library [36]. The mother wavelet functions leveraged in this study were the commonly used ones, namely, Mexican Hat, Morlet, Gaussian Derivative, Complex Gaussian Derivative, Ricker, and Shannon. Using the method [18], we determined the three most optimal mother functions for our data: for all ERG protocols, we performed CWT transform for all signals using the above mother functions. We calculated the balanced classification accuracies on the test subsets and got the top three functions for the new concatenated pediatric with adult dataset: Ricker, Gaussian, and Morlet.

To increase the efficiency [37], we use a stack of three wavelets as a 3-dimensional input image. This principle is illustrated in Figure 4. The stack can be thought of as allowing networks to extract features from different signal representations since the features can be clearly expressed for one or another continuous function.

### 4.3. Visual Transformer

Transformers have emerged as one of the most preferred models in image classification tasks, which can be primarily attributed to their computational efficiency and scalability. Figure 5 illustrates a model architecture that processes 2D wavelet data by transforming it into sequences of flattened 2D patches. These patches undergo a trainable linear projection to map them into a constant latent vector size. Before processing the patches through the encoder, a learnable embedding is added at the beginning of the sequence. It is then passed through a classification head so that fine-tuning can be conducted on the image representation before it is used for classification. In order to maintain positional information, position embeddings are used, and the sequence of embedded vectors is used as input to the Transformer encoder. The Transformer encoder comprises interleaved layers of multiheaded self-attention and multilayer perceptron blocks [38].

In the current work, we use two ResNet-ViT hybrid image classification models which differ in the number of parameters and computational efficiency: ViT Small (ViT_small_r26_s32_224) and Vit Tiny (Vit_tiny_r_s16_p8_224) [39,40,41]. Both models are available at the HuggingFace “transformers” repository [42]. We chose these models based on their popularity and the expected balance between computational complexity and effectiveness in image classification. They are commonly used in a variety of computer vision tasks, and their performance has been extensively tested on benchmark datasets like ImageNet [43]. The selected models differ mainly in the number of parameters. This work compares these two models and tests the relevance of using a heavier model to improve the metrics. Model parameters are shown in Table 2.

### 4.4. Metrics

Several metrics, including Precision, Recall, and F1 Score, were calculated to analyze the model’s performance. These metrics provide a comprehensive understanding of the model’s accuracy and effectiveness:(2)$$Precision=\frac{TP}{TP+FP},$$
(3)$$Recall=\frac{TP}{TP+FN},$$
(4)$$F1\;Score=\frac{2\times Precision\times Recall}{Precision+Recall},$$
where

TP=TruePositive,FP=FalsePositive,FN=FalseNegative.

Since the test subset reflects the real-world distribution and is not balanced, we should consider Balanced Accuracy:(5)$$Balanced\;Accuracy=\frac{Sensitivity+Specificity}{2},$$
where
(6)$$Sensitivity=Recall=\frac{TP}{TP+FN},$$
(7)$$Specificity=\frac{TN}{TN+FP}.$$

## 5. Results

The experiment results are shown in Table 3 and Figure 6. Table 3 shows measures of model performance for all tested cases. The performance was measured as Balanced Accuracy (BACC), F1 Score, Precision (P), and Recall (R). Figure 6 shows the accuracy of the analyzed cases as box plots. Each box was taken for five folds. Figure 6 is related to the ViT small model; Figure 6b is related to the ViT Tiny model. The accuracy of both models is shown for the tested wavelet stack, and each mother functions independently. Figure 6 corresponds to the Maximum, Scotopic, and Photopic protocols.

Table 3 and Figure 6 illustrate the advantages of employing a combination of wavelets in comparison to using individual wavelets alone. On average, the proposed method exhibits a 7.6% higher accuracy compared to the cases where only single wavelets are utilized.

However, it is essential to acknowledge that the precision measure for Scotopic signals is lower than that achieved using individual wavelets. This phenomenon can be attributed to the fact that the Scotopic sample is the smallest, so less precision gives an estimate of more false positives. This is not so critical because we err toward the presumably sick group.

The ViT Small model demonstrates a mere 1% increase in accuracy compared to the ViT Tiny model. However, the Tiny model possesses fewer parameters (10.4 against 36.4 million) and incurs less GMAC (0.4 against 3.4). The model’s executions were tested on a local machine with AMD Ryzen 9 5900 hx × 16 processors. The execution time of ViT Small is 61.4 ms, and the execution time of ViT Tiny is 20.4 ms, which is three times faster. Hence, it is recommended as the primary solution for future research applications.

## 6. Discussion

ViT Tiny model is one suitable for real-time scale applications on terminal devices for doctors. As some justification for this state, the comparison with ViT Small model suggests that the model provides just a 1% increase in accuracy compared to the ViT Tiny model with more parameters (36.4 million against 10.4 million) and greater GFLOPS (3.4 against 0.4).

This research shows the ability to keep pediatric and adult sets together for analysis. This can help to increase the accuracy in pediatric cases where the sample size, as usual, is dramatically smaller than for adult patients.

The results demonstrate that the proposed method achieves an 88% classification accuracy for Maximum 2.0 ERG signals, 85% for Scotopic 2.0 ERG signals, and 91% for Photopic 2.0 ERG signals. These accuracy levels represent an average improvement of 7.6% compared to previous work. By combining wavelets as input to the neural network decision-making systems, the authors observe an enhanced performance in accurately classifying ERG signals, surpassing the results obtained through individual wavelets independently.

Let us also denote that the motivation of the current research lies in the two observations taken from our previous research. The first is the lack of difference in the results for pediatric and adult cases. The second is that different wavelet functions may lead to highlighting different parts of the signal. As the study shows, these notations influence the results. Also, the achievements in DL decision-making systems allow us to increase the accuracy while having computation demands small enough. The study shows that combining these factors can be applied to the considered task.

However, the limitations of the work can be found in restrictions of the selected CWT mother functions and neural network families. As was mentioned above, the selected wavelets were chosen to provide the best performance according to the previous research.

Furthermore, the study addresses the importance of wavelet selection in achieving accurate results in ERG signal analysis. The authors recommend using ViT architecture in conjunction with the Ricker, Gauss, and Mexican Hat wavelet functions for forthcoming applications. We specifically suggest employing the ViT Tiny model due to its comparable accuracy and lower computational complexity compared to the ViT Small model.

The current study also highlights the necessity for balanced datasets in achieving reliable results. To address the issue of dataset imbalance, we employed under-sampling techniques to balance the highly unbalanced ERG signal dataset used in the study. This ensures a fair representation of both healthy and unhealthy subjects in the dataset.

Future research combining wavelet analysis and DL models should explore a broader array of wavelet functions and neural network architectures. This paper focused on the Ricker, Gauss, and Mexican Hat functions in conjunction with the ViT architecture. However, further investigations may reveal other beneficial combinations. Moreover, as the research was conducted using ERG signals from a particular dataset, testing the proposed method with different datasets or real-world clinical data could help assess its robustness and generalizability.

ERG signals have already proven effective in diagnosing various conditions affecting the retina, including inherited or acquired eye diseases. The use of AI is not new in ophthalmology, and its application to full-field ERGs is already explored. Study [44] demonstrates the applicability of ML directly to full-field ERG analysis in Stargardt disease-a genetic disorder that affects the retina. Study [45] proposes a framework for the early detection of glaucoma using an ML algorithm capable of leveraging medically relevant information that ERG signals contain.

Moreover, the central nervous system (CNS) and its function can be readily accessed through the ERG [23]. By analyzing the ERG waveform, potential biomarkers can be identified for the early detection of ADHD and bipolar disorder. Researchers have applied signal analysis techniques, such as wavelets and variable frequency complex demodulation, to studies in ASD [3] and ADHD [46] to fully leverage the potential of ERG in classifying or detecting CNS disorders at an earlier stage. These initial studies have identified the potential for identifying features extracted from signal analysis to improve ML classification models. DL approaches could further enhance the accuracy of ERG signal classification, leading to improved quality of ASD detection in its early stages and better long-term outcomes for individuals with ASD.

The studies mentioned above claim accuracy ranging from 85% to 92%, and we believe that the new results of further studies should strive for these values. However, it should be noted that the performance of the models strongly depends on the dataset, and for an objective comparison of the models, they should be tested on the same data. It should also be noted that using ML and DL models is currently considered only as an aid, and the final diagnosis will still be made by medics.

## 7. Conclusions

The currently obtained results continue the previously published studies in the medical assistance system of eye disease determination project based on the ERG signals. The main idea of wavelet combining as input for neural network decision-making systems is proposed and tested for the main ERG protocols: Maximum 2.0, Scotoic 2.0, and Photopic.

Among analyzed cases, it is proposed to use Ricker, Gauss, and Mexican Hat mother wavelets functions with ViT architecture for the following research applications. The method provides 88% accuracy for Maximum 2.0 ERG, 85% accuracy for Scotopic 2.0, and 91% accuracy for Photopic 2.0 ERG signals and a balanced database. The obtained results are 7.6% more accurate than for each considered independent wavelet.

To conclude, the research paper has made significant contributions to the advancement of the field of ophthalmology through the innovative application of wavelet analysis combined with DL techniques for more accurate classification of ERG signals. Developing an optimal decision system based on these methods is a notable contribution with essential implications for more effective diagnosis and treatment of retinal diseases. Thus, the findings of this study provide valuable insights not only for the discipline of ophthalmology but also for implementing such analytical approaches in other electrophysiological domains that warrant precise signal classification.

## Figures and Tables

**Figure 1 sensors-23-08727-f001:**
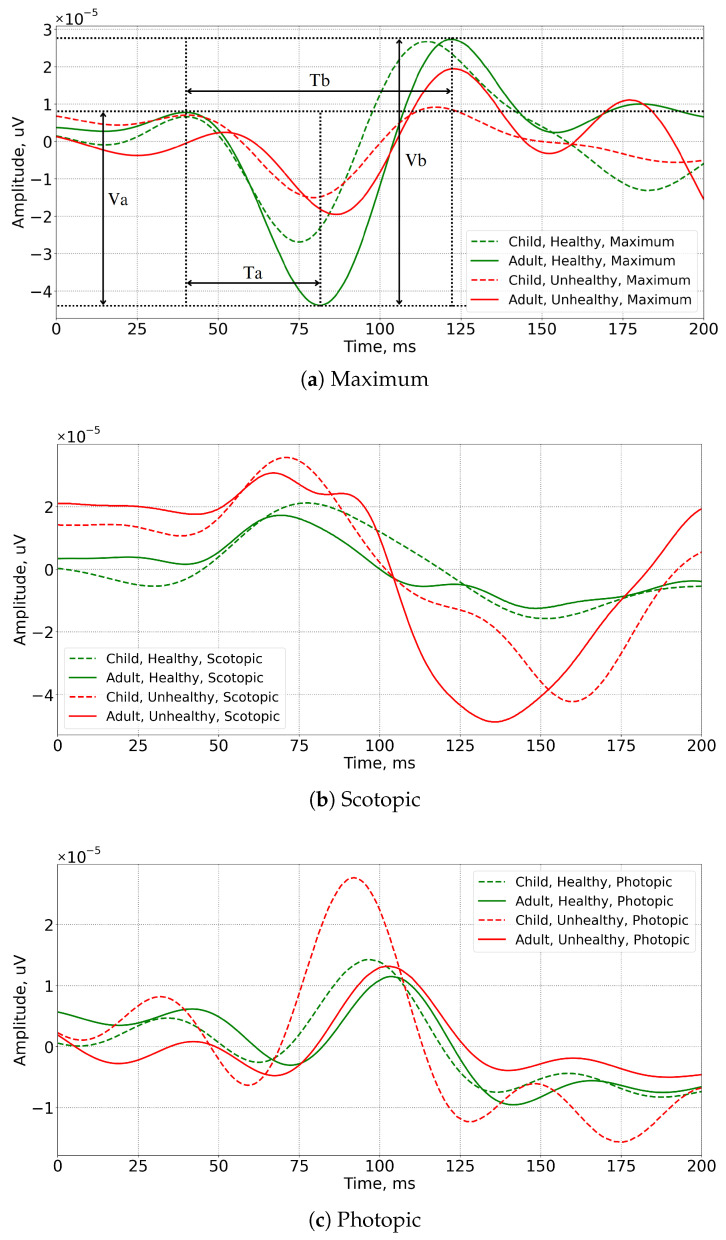
Illustration of Maximum (**a**), Scotopic (**b**), and Photopic (**c**) ERG signals: green and red lines represent healthy and unhealthy subjects; solid and dashed lines represent adult and pediatric signals, respectively.

**Figure 2 sensors-23-08727-f002:**
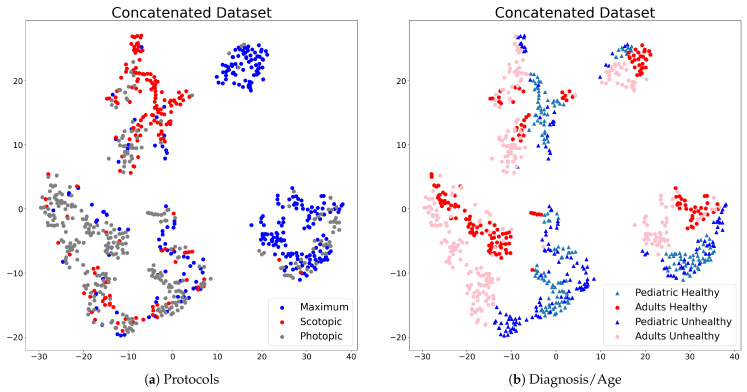
Visualization of the dataset: (**a**) Protocols: Maximum ERG Response (blue), Scotopic ERG Response (red), and Photopic ERG Response (gray); (**b**) healthy (green and red) and unhealthy (pink and blue) subjects for both pediatric (triangle) and adult (circle) cases.

**Figure 3 sensors-23-08727-f003:**
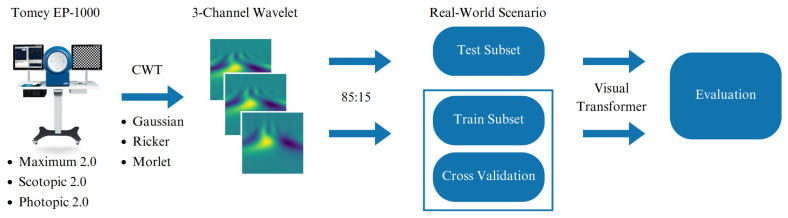
Pipeline of the experiments.

**Figure 4 sensors-23-08727-f004:**
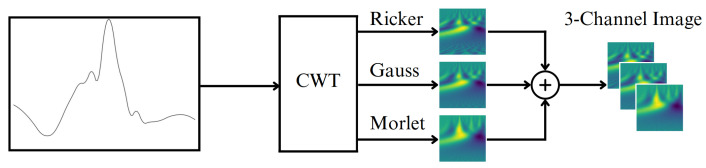
Illustration of stack of the optimal wavelet combination.

**Figure 5 sensors-23-08727-f005:**
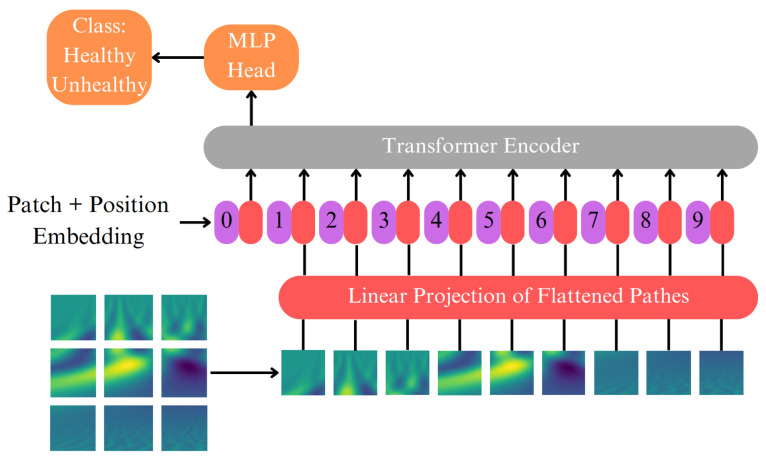
Illustration of the ViT general structure.

**Figure 6 sensors-23-08727-f006:**
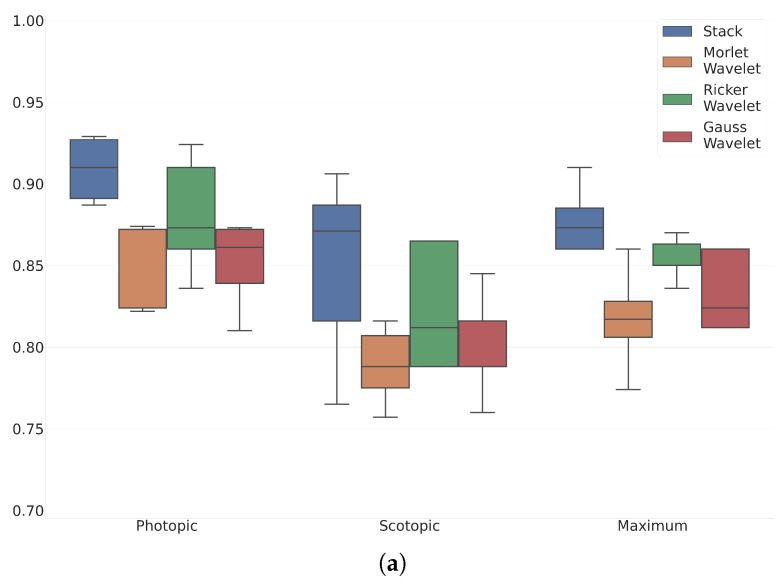
Accuracy box plots of the analyzed models for ViT Small (**a**) and ViT Tiny (**b**) models. The accuracy of both models is shown for the tested wavelet stack, and each mother functions independently.

**Table 1 sensors-23-08727-t001:** Dataset entries before and after balancing for adult, pediatric, and merged signals sets.

Pediatric				Adult				Merged	
**Unbalanced Dataset**		**Balanced Dataset**		**Unbalanced Dataset**		**Balanced Dataset**		**Balanced Dataset**	
healthy	unhealthy	healthy	unhealthy	healthy	unhealthy	healthy	unhealthy	healthy	unhealthy
Maximum 2.0 ERG Response									
143	60	62	60	148	66	102	66	164	126
Scotopic 2.0 ERG Response									
52	48	52	48	104	33	51	33	103	81
Photopic 2.0 ERG Response									
171	68	68	63	228	86	134	86	202	149

**Table 2 sensors-23-08727-t002:** Model properties.

	ViT Small	ViT Tiny
GFLOPS	3.5	0.4
parameter number (M)	36.4	10.4
activations (M)	9.4	1.9
image size	224 × 224	224 × 224
backbone	ResNet	ResNet
embed_dim	384	192
num_heads	6	3
depth	12	12
pretrain	ImageNet-21k	ImageNet-21k

**Table 3 sensors-23-08727-t003:** Experiment results.

		ViT Small				ViT Tiny			
**Wavelet**	**prt**	**BACC**	**F1**	**R**	**P**	**BACC**	**F1**	**R**	**P**
stack	**Maximum**	**0.88**	**0.87**	**0.84**	**0.89**	**0.87**	**0.86**	**0.83**	**0.88**
morl		0.82	0.79	0.79	0.81	0.80	0.78	0.77	0.79
gaus8		0.83	0.82	0.79	0.85	0.83	0.81	0.79	0.84
mexh		0.85	0.83	**0.84**	0.82	0.84	0.82	0.83	0.81
stack	**Scotopic**	**0.85**	**0.80**	**0.83**	0.77	**0.83**	**0.77**	**0.81**	0.75
morl		0.79	0.74	0.69	0.81	0.79	0.75	0.70	**0.81**
gaus8		0.81	0.77	0.73	0.81	0.77	0.73	0.69	0.77
mexh		0.82	0.79	0.76	**0.83**	0.80	0.76	0.72	0.80
stack	**Photopic**	**0.91**	**0.90**	**0.91**	**0.88**	**0.90**	**0.88**	**0.90**	**0.87**
morl		0.84	0.83	0.81	0.85	0.83	0.81	0.79	0.83
gaus8		0.85	0.83	0.84	0.82	0.84	0.82	0.83	0.81
mexh		0.88	0.87	0.86	0.88	0.88	0.86	0.85	**0.87**

## Data Availability

Zhdanov, A.E.; Dolganov, A.Y.; Borisov, V.I.; Lucian, E.; Bao, X.; Kazaijkin, V.N.; Ponomarev, V.O.; Lizunov, A.V.; Ivliev, S.A. 355 OculusGraphy: Pediatric and Adults Electroretinograms Database, 2020. https://doi.org/10.21227/y0fh-5v04, accessed on 19 September 2023.

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
