# Peer review of "Enhancing Electroretinogram Classification with Multi-Wavelet Analysis and Visual Transformer"

_sensors, 2023, doi:10.3390/s23218727_

Round 1

Reviewer 1 Report

The authors researched improvements in Electroretinogram (ERG) classification by applying multi-wavelets and visual transformer. For the experiment, they used an adult and pediatric recorded ERG signal. The CWT has been applied to data sets. Also, the visual transformer-based architectures were tested on a time-frequency representation of the signals. The results show that the proposed method achieved 88% classification accuracy for maximum 2.0 ERG, 85% for scotopic 2.0, and 91% for photopic 2.0 protocols. They claim that, on average, the results are improved by 7.6% compared to previous work.

This is an interesting research report. In my opinion, the article is professionally executed. The manuscript is technically well-written. It consists of seven chapters: 1. Introduction, 2. Related Works, 3. Dataset Investigation, 4. Method, 5 Results, 6. Discussion, and 7. Conclusions. The experimental protocol seems repeatable. The result seems valid.

Before the publication, I have the following comments/concerns for the authors to address:

  1. The sentences in L15 and L19 are similar. Please correct it.
  2. Please add the equipment description used in the experiment (PC or Mac configuration; CPU, DRAM, graphical card, software, etc., ...). 
  3. L181: Please consider explaining the method of choosing the optimal wavelets in a sentence or two. In my opinion, only pointing to the reference [37] is not enough. It will add to the manuscript's readability.
  4. Please consider showing Figure 5 at the beginning of chapter four. It represents the proposed model. It would add to the manuscript's readability. 
  5. Do you have execution times for the proposed networks (ViT Small and ViT Tiny)?

Author Response

Authors thanks reviewer for the comments.
Comments/concerns for the authors to address:

  1. The sentences in L15 and L19 are similar. Please correct it. 
  2. Please add the equipment description used in the experiment (PC or Mac configuration; CPU, DRAM, graphical card, software, etc.). 
  3. L181: Please consider explaining the method of choosing the optimal wavelets in a sentence or two. In my opinion, only pointing to the reference [37] is not enough. It will add to the manuscript's readability.
  4. Please consider showing Figure 5 at the beginning of chapter four. It represents the proposed model. It would add to the manuscript's readability. 
  5. Do you have execution times for the proposed networks (ViT Small and ViT Tiny)?

Reply:

  1. Thank you for the comment! We corrected the paragraph.
  2. Thank you for the comment! We added this information here: ADAM optimization with an initial learning rate of 0.001 was employed during training. Each model was trained until convergence using early stopping criteria based on the validation loss. A batch size of 16 was used, and training was performed on a single NVIDIA V100 graphics processing unit on a machine with two Intel Xeon Gold 6134 3.2 GHz and 96 GB RAM.
  3. Thank you for the comment! We added this information here: The mother wavelet functions leveraged in this study were the commonly used ones, namely Mexican Hat, Morlet, Gaussian Derivative, Complex Gaussian Derivative, Ricker, and Shannon. Using the method, we determined the three most optimal mother functions for our data: for all ERG protocols, we performed CWT transform for all signals using the above mother functions. We calculated the balanced classification accuracies on the test subsets and got the top three functions for the new concatenated children with adults dataset: Ricker, Gaussian, and Morlet.
  4. Thank you for the advice! We changed the order of the subsections.
  5. Thank you for the question! We added this information here: The model's executions were tested on a local machine with AMD Ryzen 9 5900hx × 16 processors. The execution time of ViT Small is 61.4 ms, and the execution time of ViT Tiny is 20.4 ms, which is three times faster.

Reviewer 2 Report

The work by Kulyabin et al. is based too much on their previous work published the same year in the same journal. The results and discussion are too short and do not justify this work. The significance of this work is also on the low side. Therefore, I cannot recommend acceptance of this work in the current state.

Overall, the language quality is fine

Author Response

The work by Kulyabin et al. is based too much on their previous work published the same year in the same journal. The results and discussion are too short and do not justify this work. The significance of this work is also on the low side. Therefore, I cannot recommend acceptance of this work in the current state.

Reply:

Thank you for the comment!

In this work, we show the possibility of simultaneous analysis of adult and pediatric signals, therefore increasing the dataset. Unlike previous work, we use different models of Transformers: expanded dataset and wavelet concatenation method allow us to improve the proposed method's accuracy significantly.

Below are the differences between the previous and present works in detail:

1 - In this work, we show the possibility of simultaneous analysis of adult and pediatric signals, therefore increasing the dataset.

Note that in our previous work, signals from only one group - pediatric - were used to train neural networks for different protocols (Maximum, Scotopic, Photopic 2.0 ERG Responses) In previous ophthalmology papers, researchers separated pediatric and adult signals, considering the differences between them [1, 2] and contrasts in conducting the ERG experiments [3, 4, 5]. However, to increase the sample size, assuming that a larger number of samples is required for Transformer training, the present study combines pediatric and adult signals from the same protocol, which is consistent with the international standard [6]. 

A cluster analysis of the dataset was performed, and its features were described in the text: "Fig. 2b results show that the adult and pediatric signals could be considered to be processed together due to the high mixing among them in each signal type. According to the distribution shown in Fig. 2, the intragroup scatter of parameters matches the intergroup scatter between pediatric and adults. As a result of this reasoning, it is possible to conduct a joint analysis of healthy and unhealthy subjects belonging to different age groups."

2 - Unlike previous work, we use different transformer models: expanded dataset and wavelet concatenation method allow us to improve the proposed method's accuracy significantly. Since the results of previous work show an influential superiority of transformers, this article uses ViT models. Also, the choice of transformers is motivated in the current work. We use two ResNet - ViT hybrid image classification models, which differ in the number of parameters and computational efficiency: ViT Small and Vit Tiny [7-9]. Both models are available at the HuggingFace repository [10]. We chose these models based on their popularity and the expected balance between computational complexity and effectiveness in image classification. They are commonly used in various computer vision tasks, and their performance has been extensively tested on benchmark datasets like ImageNet. This work compares these two models and tests the relevance of using a heavier model to improve the metrics.

3 - A stack of wavelets was used for the first time for ERG signals.

Previous research [11] shows that the use of wavelet analysis potentially expands the possibilities of diagnosing diseases using ERG. In particular, it allows us to describe the slightest frequency and temporal differences from the signals of healthy subjects. Previous research shows that the basis functions used in this study are the most effective for the protocol used. However, a comprehensive assessment of these mother functions would allow for choosing the optimal one. Using a stack will enable us to solve this problem, which was solved earlier in the other domain [12].

We also enlarged the discussion section according to the reviewer's comments.

[1]  Wang, J., Wang, Y., Guan, W. et al. Full-field electroretinogram recorded with skin electrodes in 6- to 12-year-old children. Doc Ophthalmol (2023). https://doi.org/10.1007/s10633-023-09944-9

[2]  Luu, C. D., Lau, A. M., & Lee, S. Y. (2006). Multifocal electroretinogram in adults and children with myopia. Archives of Ophthalmology, 124(3), 328-334.

[3]  Marmoy, O. R., Moinuddin, M., & Thompson, D. A. (2022). An alternative electroretinography protocol for children: a study of diagnostic agreement and accuracy relative to ISCEV standard electroretinograms. Acta Ophthalmologica, 100(3), 322-330.

[4]  Westall, C. A., Panton, C. M., & Levin, A. V. (1998). Time courses for maturation of electroretinogram responses from infancy to adulthood. Documenta ophthalmologica, 96, 355-379.

[5]  Wongpichedchai, S., Hansen, R. M., Koka, B., Gudas, V. M., & Fulton, A. B. (1992). Effects of halothane on children's electroretinograms. Ophthalmology, 99(8), 1309-1312.

[6] Robson, A.G.; Frishman, L.J.; Grigg, J.; Hamilton, R.; Jeffrey, B.G.; Kondo, M.; Li, S.; McCulloch, D.L. ISCEV Standard for full-field clinical electroretinography (2022 update). Documenta Ophthalmologica 2022, pp. 165–177.

[7] Dosovitskiy, A.; Beyer, L.; Kolesnikov, A.; Weissenborn, D.; Zhai, X.; Unterthiner, T.; Dehghani, M.; Minderer, M.; Heigold, G.; Gelly, S.; et al. An image is worth 16x16 words: Transformers for image recognition at scale. arXiv preprint arXiv:2010.11929 2020.

[8] Khan, S.; Naseer, M.; Hayat, M.; Zamir, S.W.; Khan, F.S.; Shah, M. Transformers in vision: A survey. ACM Computing Surveys (CSUR) 2021. 

[9] Wu, K.; Zhang, J.; Peng, H.; Liu, M.; Xiao, B.; Fu, J.; Yuan, L. Tinyvit: Fast pretraining distillation for small vision transformers. In Proceedings of the European Conference on Computer Vision. Springer, 2022, pp. 68–85.

[10] Wolf, T.; Debut, L.; Sanh, V.; Chaumond, J.; Delangue, C.; Moi, A.; Cistac, P.; Rault, T.; Louf, R.; Funtowicz, M.; et al. Transformers: State-of-the-art natural language processing. In Proceedings of the Proceedings of the 2020 conference on empirical methods in natural language processing: system demonstrations, 2020, pp. 38–45.

[11] Zhdanov, A.; Constable, P.; Manjur, S.M.; Dolganov, A.; Posada-Quintero, H.F.; Lizunov, A. OculusGraphy: Signal Analysis of the Electroretinogram in a Rabbit Model of Endophthalmitis Using Discrete and Continuous Wavelet Transforms. Bioengineering 2023, 10, 708.

[12] Arias-Vergara, T.; Klumpp, P.; Vasquez-Correa, J.; Nöth, E.; Orozco-Arroyave, J.; Schuster, M. Multi-channel spectrograms for speech processing applications using deep learning methods. Pattern Analysis and Applications 2020, pp. 1–9

Reviewer 3 Report

The paper introduced a new method to treat the signal. It provided some new techniques for this subject. However, some contents need to be enhanced.

1.     In Figure 1, please compare the healthy and unhealthy of a child or adult at the same figure to illustrate the difference.

2.     Please enlarge figure 2b or separate it into two figures, pediatric of two cases (healthy and unhealthy), and adult of two cases.

3.     Line 163-164 “In our analysis, we deliberately chose 13 as the number of nearest neighbors to attain the desired class balance.”. Please explain the number “13”, what is the criterion for selecting this number?

4.     Line 166-167, “Table 1 presents the distribution of healthy and unhealthy subjects within a balanced dataset. In this work, we balance dataset for training experiments.”

How to obtain these data? What is the source of these data sets? How to ensure these data sets could represent the actual cases?

5.     Line 249-251, “Table 3 and Figs. 6 illustrate the advantages of employing a combination of wavelets in comparison to using individual wavelets alone. On average, the proposed method exhibits a 7.6% higher accuracy compared to the cases where only single wavelets are utilized.”

What is the required accuracy of the different retinal diseases and disorders? Is it meaning for the clinical case? Please illustrate the actual case in the medical treatment

Minor editing of English language required

Author Response

The paper introduced a new method to treat the signal. It provided some new techniques for this subject. However, some contents need to be enhanced.

  1. In Figure 1, please compare the healthy and unhealthy of a child or adult at the same figure to illustrate the difference.
  2. Please enlarge figure 2b or separate it into two figures, pediatric of two cases (healthy and unhealthy), and adult of two cases.
  3. Line 163-164 “In our analysis, we deliberately chose 13 as the number of nearest neighbors to attain the desired class balance.”. Please explain the number “13”, what is the criterion for selecting this number?
  4. Line 166-167, “Table 1 presents the distribution of healthy and unhealthy subjects within a balanced dataset. In this work, we balance the dataset for training experiments.”

How to obtain these data? What is the source of these data sets? How to ensure these data sets could represent the actual cases?

  1. Line 249-251, “Table 3 and Figs. 6 illustrate the advantages of employing a combination of wavelets in comparison to using individual wavelets alone. On average, the proposed method exhibits a 7.6% higher accuracy compared to the cases where only single wavelets are utilized.”

What is the required accuracy of the different retinal diseases and disorders? Is it meaning for the clinical case? Please illustrate the actual case in the medical treatment

Reply:

  1. Thank you for the suggestion! We updated the figures.
  2. Thank you for the advice, we tried to enlarge the figures. The idea was to show the sample distribution on the one graph.
  3. Thank you for the question! We added this part: AllKNN method has a hyperparameter that affects the results of the under-sampling procedure: setting it too low or too high could lead to either removing too much of the data or removing too few data. The goal of the under-sampling in this study is to ensure a balance between healthy and unhealthy groups. For that, an array of possible numbers was selected. In this case, for Maximum and Photopic signals, we have chosen empirically to use 13 as the number of nearest neighbors to achieve the desired class balance. It is worth mentioning that the Scotopic signals were inherently balanced and did not necessitate any under-sampling technique to maintain class equilibrium.
  4. Thank you for the question! We added this part: In this work, we balanced the dataset for the training experiments. For the testing, we keep the “real-world scenario” distribution of the healthy and unhealthy patients, as the number of the patients with eye diseases is always higher on the clinic tests. (Real-world ratio of healthy and unhealthy individuals we have from the clinic. It equals to the distribution of the pure dataset. We use the signals from the publicly available dataset, which is mentioned in the “Data Availability Statement”.)
  5. ERG signals have already proven effective in diagnosing various conditions affecting the retina, including inherited or acquired eye diseases. The use of AI is not new in ophthalmology, and its application to full-field ERGs is already explored. Study [1] demonstrates the applicability of machine learning directly to full-field ERG analysis in Stargardt disease - a genetic disorder that affects the retina. Study [2] proposes a framework for the early detection of glaucoma using an ML algorithm capable of leveraging medically relevant information that ERG signals contain.

Moreover, the central nervous system (CNS) and its function can be readily accessed through the ERG [3]. By analyzing the ERG waveform, potential biomarkers can be identified for the early detection of Attention Deficit Hyperactivity Disorder (ADHD) and bipolar disorder. Researchers have applied signal analysis techniques, such as wavelets and variable frequency complex demodulation, to studies in ASD[4] and ADHD[5] to fully leverage the potential of ERG in classifying or detecting CNS disorders at an earlier stage. These initial studies have identified the potential for identifying features extracted from signal analysis to improve machine learning (ML) classification models. Deep learning (DL) approaches could further enhance the accuracy of ERG signal classification, leading to improved quality of ASD detection in its early stages and better long-term outcomes for individuals with ASD.

The studies mentioned above claim accuracy ranging from 85 to 92%, and we believe that the new results of further studies should strive for these values. However, it should be noted that the performance of the models strongly depends on the dataset, and for an objective comparison of the models, they should be tested on the same data. It should also be noted that using ML and DL models is currently considered only as an aid, and the final diagnosis will still be made by medics.

[1] Sophie L. Glinton, Antonio Calcagni, Watjana Lilaonitkul, Nikolas Pontikos, Sandra Vermeirsch, Gongyu Zhang, Gavin Arno, Siegfried K. Wagner, Michel  Michaelides, Pearse A. Keane, Andrew R. Webster, Omar A. Mahroo, Anthony G. Robson; Phenotyping of ABCA4 Retinopathy by Machine Learning Analysis of Full-Field Electroretinography. Trans. Vis. Sci. Tech. 2022;11(9):34. https://doi.org/10.1167/tvst.11.9.34.

[2] Gajendran, M. K., Rohowetz, L. J., Koulen, P., & Mehdizadeh, A. (2022). Novel machine-learning based framework using electroretinography data for the detection of early-stage glaucoma. Frontiers in Neuroscience, 16, 869137.

[3] P. A. Constable, F. Marmolejo-Ramos, M. Gauthier, I. O. Lee, D. H. Skuse, and D. A. Thompson, “Discrete wavelet transform analysis of the electroretinogram in autism spectrum disorder and attention deficit hyperactivity disorder,” Frontiers in Neuroscience, vol. 16, p. 890461, 2022.

[4] S. M. Manjur, M.-B. Hossain, P. A. Constable, D. A. Thompson, F. Marmolejo-Ramos, I. O. Lee, D. H. Skuse, and H. F. Posada-Quintero, “Detecting autism spectrum disorder using spectral analysis of electroretinogram and machine learning: Preliminary results,” in 2022 44th Annual International Conference of the IEEE Engineering in Medicine & Biology Society (EMBC). IEEE, 2022, pp. 3435–3438.

[5] Paul A. Constable, Jeremiah K. Lim, and Dorothy A. Thompson, “Retinal Electrophysiology in Central Nervous System Disorders. A review of human and mouse studies,” Frontiers in Neuroscience, vol. 17, 2023.

Round 2

Reviewer 3 Report

All the problems have been solved adequately.

Minor editing of English language required